# Intratumoral IL-12 delivery empowers CAR-T cell immunotherapy in a pre-clinical model of glioblastoma

Giulia Agliardi[1,6], Anna Rita Liuzzi[2,6], Alastair Hotblack [1], Donatella De Feo[2], Nicolás Núñez [2], Cassandra L. Stowe[1], Ekaterina Friebel[2], Francesco Nannini [1], Lukas Rindlisbacher[2], Thomas A. Roberts [3], Rajiv Ramasawmy[3], Iwan P. Williams[1], Bernard M. Siow [3,4], Mark F. Lythgoe[3], Tammy L. Kalber [3], Sergio A. Quezada [1], Martin A. Pule[1], Sonia Tugues[2], Karin Straathof [1,5,7✉] & Burkhard Becher [2,7✉]

Glioblastoma multiforme (GBM) is the most common and aggressive form of primary brain cancer, for which effective therapies are urgently needed. Chimeric antigen receptor (CAR)-based immunotherapy represents a promising therapeutic approach, but it is often impeded by highly immunosuppressive tumor microenvironments (TME). Here, in an immuno-competent, orthotopic GBM mouse model, we show that CAR-T cells targeting tumor-specific epidermal growth factor receptor variant III (EGFRvIII) alone fail to control fully established tumors but, when combined with a single, locally delivered dose of IL-12, achieve durable anti-tumor responses. IL-12 not only boosts cytotoxicity of CAR-T cells, but also reshapes the TME, driving increased infiltration of proinflammatory CD4$^+$ T cells, decreased numbers of regulatory T cells (Treg), and activation of the myeloid compartment. Importantly, the immunotherapy-enabling benefits of IL-12 are achieved with minimal systemic effects. Our findings thus show that local delivery of IL-12 may be an effective adjuvant for CAR-T cell therapy for GBM.

[1] Research Department of Hematology, Cancer Institute, University College London, Paul O'Gorman Building, WC1E 6DD London, UK. [2] Institute of Experimental Immunology, University of Zurich, 8057 Zurich, Switzerland. [3] Centre for Advanced Biomedical Imaging (CABI), University College London, Paul O'Gorman Building, WC1E 6DD London, UK. [4] The Francis Crick Institute, NW1 1AT London, UK. [5] UCL Great Ormond Street Institute of Child Health Biomedical Research Centre, WC1N 1EH London, UK. [6] These authors contributed equally: Giulia Agliardi, Anna Rita Liuzzi. [7] These authors jointly supervised this work: Karin Straathof, Burkhard Becher. ✉email: k.straathof@ucl.ac.uk; becher@immunology.uzh.ch

Glioblastoma multiforme (GBM) is the most common primary malignant brain tumor in adults, accounting for ~60–70% of gliomas[1]. Standard therapy consists of tumor resection followed by radiotherapy and concomitant temozolomide. Due to the infiltrative nature of these tumors, recurrence either at the margins of the original resection or at distant structures of brain parenchyma occurs in most cases[2]. As a result, the patient outcome is dismal with a median survival of 14.6 months and an average 5-year survival rate of <5%[3,4].

Treatment with T cells redirected to tumor specificity with a chimeric antigen receptor (CAR) may be well suited to treat intracranial tumors due to the ability of T cells to access the central nervous system (CNS) and migrate to infiltrative sites of disease. Treatment with CD19-directed CAR-T cells is effective not only against leptomeningeal leukemic infiltrates[5] but also brain parenchymal B-cell lymphomatous deposits[6].

In adult GBM, a case report of local and distant eradication of intracranial and spinal deposits of GBM following intraventricular infusion of IL13Rα2-CAR-T cells indicates the potential of this approach in GBM[7,8]. However, in contrast to the sustained complete remissions observed in hematological malignancies[6], in the majority of patients with GBM CAR-T cell therapy has not resulted in clinical benefit[9,10]. Tumor heterogeneity and antigen loss as well as the highly immune inhibitory tumor microenvironment are likely the key barriers to achieving durable anti-tumor immunity.

CAR-T cell efficacy is impaired by an adaptive immune suppressive response, i.e. upregulation of immune inhibitory molecules, such as programmed cell death ligand-1 (PD-L1), indoleamine-2,3-deoxygenase 1 (IDO1), and infiltration of $T_{regs}$[10,11]. These results highlight the need for additional therapeutic strategies to counteract the hostile TME and overcome tumor heterogeneity.

IL-12, a pro-inflammatory cytokine with potent tumor-suppressive activity, represents a promising candidate for combinatorial immunotherapy. IL-12 can directly support persistent cytotoxic activity of T cells, as well as improve antigen presentation, mitigate against antigen-negative escape, and reshape endogenous immune inhibitory cells within the TME[12,13]. Combination of CAR-T cells with IL-12 has been shown to enhance anti-tumor response in mouse models of extracranial tumors including leukemia and ovarian cancer[14–17]. However, systemic IL-12 is poorly tolerated[18], while delivery via engineered T cells is also associated with severe toxicity[19]. Hence, a delivery method which achieves the benefits of IL-12 in the TME without systemic toxicity is desirable[13,20].

Here we propose the use of a single intratumoral dose of recombinant single chain IL-12 fused to the Fc portion of murine IgG3 (hereafter called IL-12:Fc) in combination with systemic CAR-T cell therapy. In a syngeneic mouse model of GBM, we show that this combinatorial treatment results in complete eradication of established gliomas and demonstrate the effects of IL-12 on EGFRvIII-specific CAR-T cell fitness and reshaping of the TME that underpin achieved anti-tumor immunity. This localized delivery of IL-12 resulted in only mild systemic effects. Our results demonstrate that local administration of IL-12 may overcome barriers encountered by CAR-T cell therapy for GBM and provide a rationale for a combination treatment approach in clinical study.

## Results

### EGFRvIII-specific murine CAR-T cells fail to control large established gliomas.
We used an immunocompetent orthotopic GL261 mouse model of glioma to evaluate the anti-tumor activity of EGFRvIII-specific murine CAR-T cells. Parental GL261 line was transduced with the extracellular portion of EGFR, containing the variant III mutation fused with the transmembrane domain of the murine EGFR (Fig. 1A), which represents a common mutation in human GBM[21]. A second generation CAR was constructed using the single chain variable fragment derived from EGFRvIII-specific MR1 antibody[22], murine CD28-derived transmembrane domain and murine CD28 and CD3ζ intracellular domains. Truncated murine CD34 was co-expressed with the CAR to allow detection of CAR-transduced T cells (Fig. 1B; Supplementary Fig. 1A). Murine T cells transduced to express the EGFRvIII-specific CAR efficiently killed EGFRvIII[+] GL261 but not parental GL261 in vitro (Supplementary Fig. 1B).

Next, we tested efficacy of EGFRvIII-directed CAR-T cells in vivo in mice bearing orthotopic EGFRvIII[+] GL261 tumors. Tumor growth was monitored using a 1 Tesla magnetic resonance imaging (1T-MRI) system. When tumors were visible by MRI (day 10 post implantation), mice received 5 Gy total body irradiation (TBI) as preparative lymphodepletion. Preparative lymphodepletion is required to achieve engraftment and anti-tumor efficacy of adoptively transferred T cells and is now a standard component of CAR-T cell therapy[23]. TBI was followed by intravenous injection of $2.5 \times 10^6$ CAR-T cells (Supplementary Fig. 1C). By day 7 post transfer, CAR-T cells infiltrated tumors (Supplementary Fig. 1D, E) as shown by immunohistochemistry for CD34. Tumor infiltration was antigen-specific, as infusion of CAR-T cells specific for human CD19 (used as negative control) did not accumulate within the tumor (Supplementary Fig. 1F, G). In this setting, CAR-T cell administration resulted in tumor clearance and long-term survival in 50% of treated mice ($p = 0.0028$) (Supplementary Fig. 1H), demonstrating their effectiveness when administered at early stages after tumor inoculation.

To recapitulate the clinical observation that treatment with CAR-T cells alone is not sufficient to eradicate glioblastoma, we developed a 'stress' model using a reduced dose of CAR-T cells ($1 \times 10^6$) administered at day 11 (early stage) or day 17 (late stage) post tumor implantation (Fig. 1C). Early administration of CAR-T cells resulted in control of tumor growth and significantly improved survival (Fig. 1D–F). In contrast, late administration of CAR-T cells resulted in a poor tumor control and subsequent tumor outgrowth without improved survival (Fig. 1G–I). Taken together, these data suggest that CAR-T cells can only control newly established tumors but fail to control larger ones, a condition that more closely resembles the stages at which CAR-T cells will be used in the clinic.

### Local delivery of a single dose of IL-12 improves efficacy of CAR-T cells.
To boost CAR-T cells anti-tumor activity in large established gliomas, we designed a combinatorial immunotherapy approach. We evaluated whether administration of IL-12:Fc (Fig. 2A) in combination with CAR-T cells would improve tumor control in late stage EGFRvIII[+] GL261 gliomas. Mice received 5 Gy TBI on day 15 post tumor implantation, followed by a single intra-tumoral injection (300 ng) of IL-12:Fc on day 20 (Fig. 2B). Mice were then treated with $1 \times 10^6$ EGFRvIII-directed CAR-T or non-transduced (NT) T cells administered intravenously on day 21 (Fig. 2B). Tumor volumes as measured by MRI showed that, while both single therapies (CAR+PBS or NT+IL-12:Fc) were only able to delay tumor growth, the combination of systemic EGFRvIII-specific CAR infusion and local IL-12:Fc administration eliminated tumors in most treated mice and showed a synergistic effect on the overall survival (Fig. 2C–E). Next, we assessed the effect of IL-12:Fc on CAR-T efficacy in a second intracranial tumor model. Here, we used intracranial implantation of B16.F10 cells to recapitulate brain metastases of melanoma[20,24,25]. B16.F10 were transduced to express GD2, a tumor antigen expressed on tumors of neuroectodermal origin

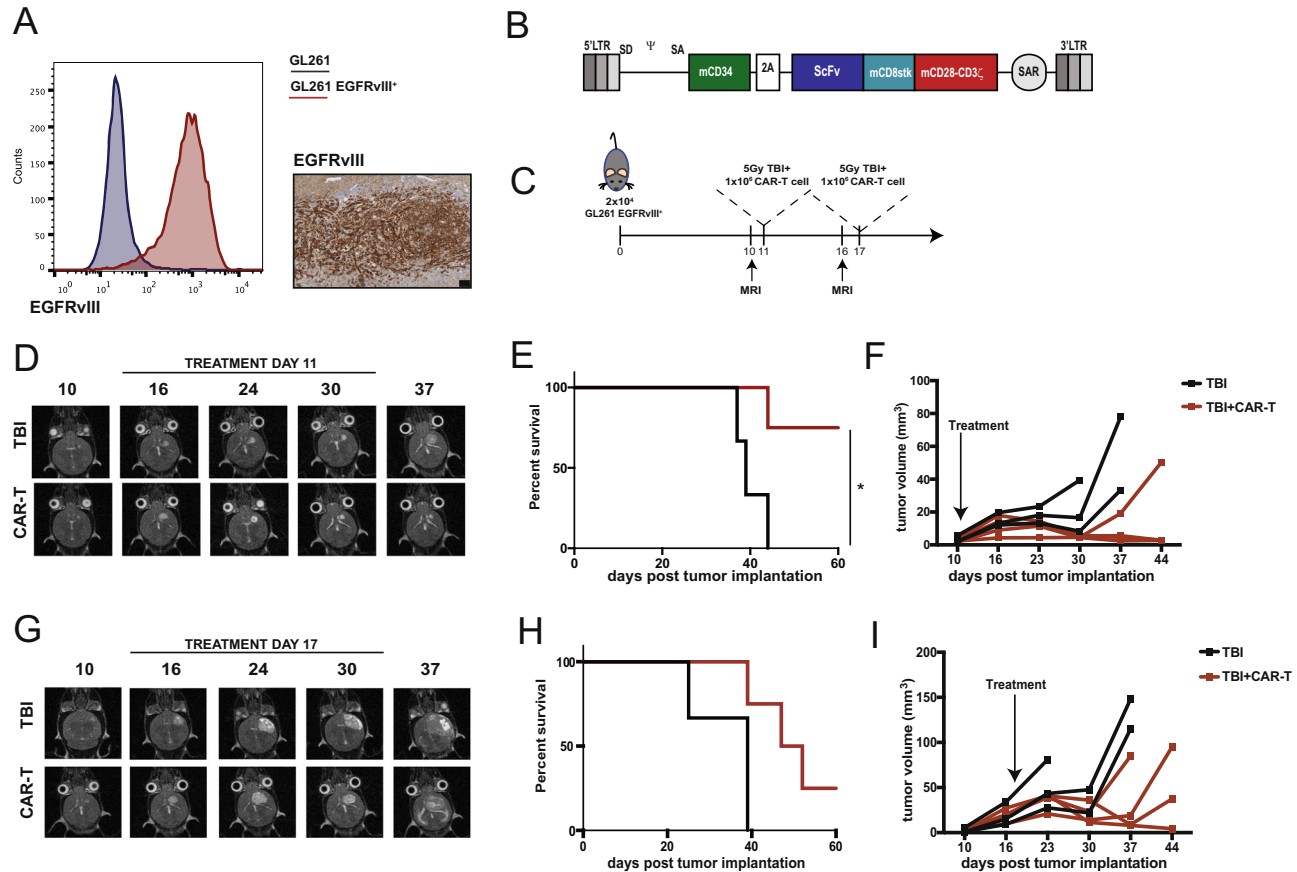

**Fig. 1 EGFRvIII-specific CAR-T cells control newly established orthotopic tumors but fail to control large ones. A** GL261 were transduced with a retroviral vector to express the murine version of EGFRvIII. The mutated portion of EGFRvIII was fused with the transmembrane domain of the mouse EGFR to obtain cells which express the epitope on the surface. Left panel: flow cytometry staining of wild type GL261 (blue) and GL261 transduced to express EGFRvIII (red). Right panel: immunohistochemistry staining for EGFRvIII on orthotopically implanted tumors (scale bar represents 100 μm). One representative tumor is shown of four mice. **B** Murine CAR construct. The C-terminal portion of murine CD34 was included as marker gene and separated by a T2A peptide from the CAR construct, which included an ScFv to graft specificity, a CD8 stalk, and CD28-CD3ζ as activation domains. MR1 was used as ScFv specific for EGFRvIII, while 4g7 was used as ScFv specific for human CD19, used as negative control CAR. **C** "Stress experiment". Direct comparison of the effect on tumor control of intravenous CAR-T cell administration on day 11 or 17 post tumor implantation. **D** Representative MRI images (axial orientation) of a mouse receiving either TBI only or TBI followed by CAR-T cells at day 11 post tumor implantation. **E** Survival curves ($n = 3$ for TBI, $n = 4$ for TBI+CAR mice from one experiment, $p = 0.0288$ (*), Log-rank test) and **F** tumor volume quantification. **G** Representative MRI images (axial plane) of a mouse receiving either TBI only or TBI followed by CAR-T cells at day 17 post tumor implantation. **H** Survival curves ($n = 3$ for TBI, $n = 4$ for TBI+CAR mice from one experiment) and **I** tumor volume quantification. Source data are provided as a Source Data file.

including gliomas, melanomas, and neuroblastoma[26] (Supplementary Fig. 2A). In this setting, we used a GD2-specific second generation CAR based on the single chain fragment variant derived from the K666 antibody[27]. In this highly aggressive and poorly immunogenic model[28,29], mice in the control group (NT+PBS) had a median survival of 15 days, while treatment with NT+IL-12:Fc alone did not confer a survival benefit. Of note, the administration of GD2-specific CAR-T cells improved survival over NT+PBS and NT+IL-12:Fc groups ($p = 0.01$). Moreover, combination treatment with IL-12:Fc and CAR-T cells resulted in significant improved survival as compared to either treatment alone (NT+IL-12:Fc vs. CAR+IL-12:Fc, $p < 0.0001$; CAR+PBS vs. CAR+IL-12:Fc, $p < 0.001$) (Supplementary Fig. 2B–D). These data demonstrate that combined IL-12 and CAR-T cell therapy promotes an effective and persistent anti-tumor response, even in the context of a poorly immunogenic model.

**Local IL-12 treatment reinvigorates glioma infiltrating dysfunctional CAR-T cells.** To explore the mechanism underlying the synergistic effect of the combination of CAR-T cells and IL-12,

we performed a high-parametric flow-cytometric characterization of the TME using 23 independent parameters representing lineage and activation markers (Supplementary Table 1). We selected day 8 post treatment administration (Fig. 3A), as at this time point GL261_EGFRvIII+ tumors were still comparable in size (Supplementary Fig. 3A), as well as heavily infiltrated by EGFRvIII-CAR-T cells. For this analysis, EGFRvIII-CAR-T cells were produced from CD45.1 syngeneic mice to allow discrimination from endogenous T cells. As most of the infiltrating CAR-T cells were CD8+ T cells (Supplementary Fig. 3B) CAR-T cells were defined as CD45.1+ TCR-β+CD34+CD8+ cells (Fig. 3B). We observed no differences in the number of EGFRvIII-directed CAR-T cells in brain tumors receiving local IL-12:Fc in addition to CAR-T cells, compared to those receiving CAR-T cells and PBS only (Supplementary Fig. 3C), suggesting that differences in tumor control are not due to increased tumor infiltration of adoptively transferred T cells.

We then used the representation machine-learning algorithm CellCNN[30] as an unbiased and hypothesis-free method to measure the effect of IL-12:Fc treatment on CAR-T cells. We identified a cluster of CAR-T cells, positive for the co-inhibitory

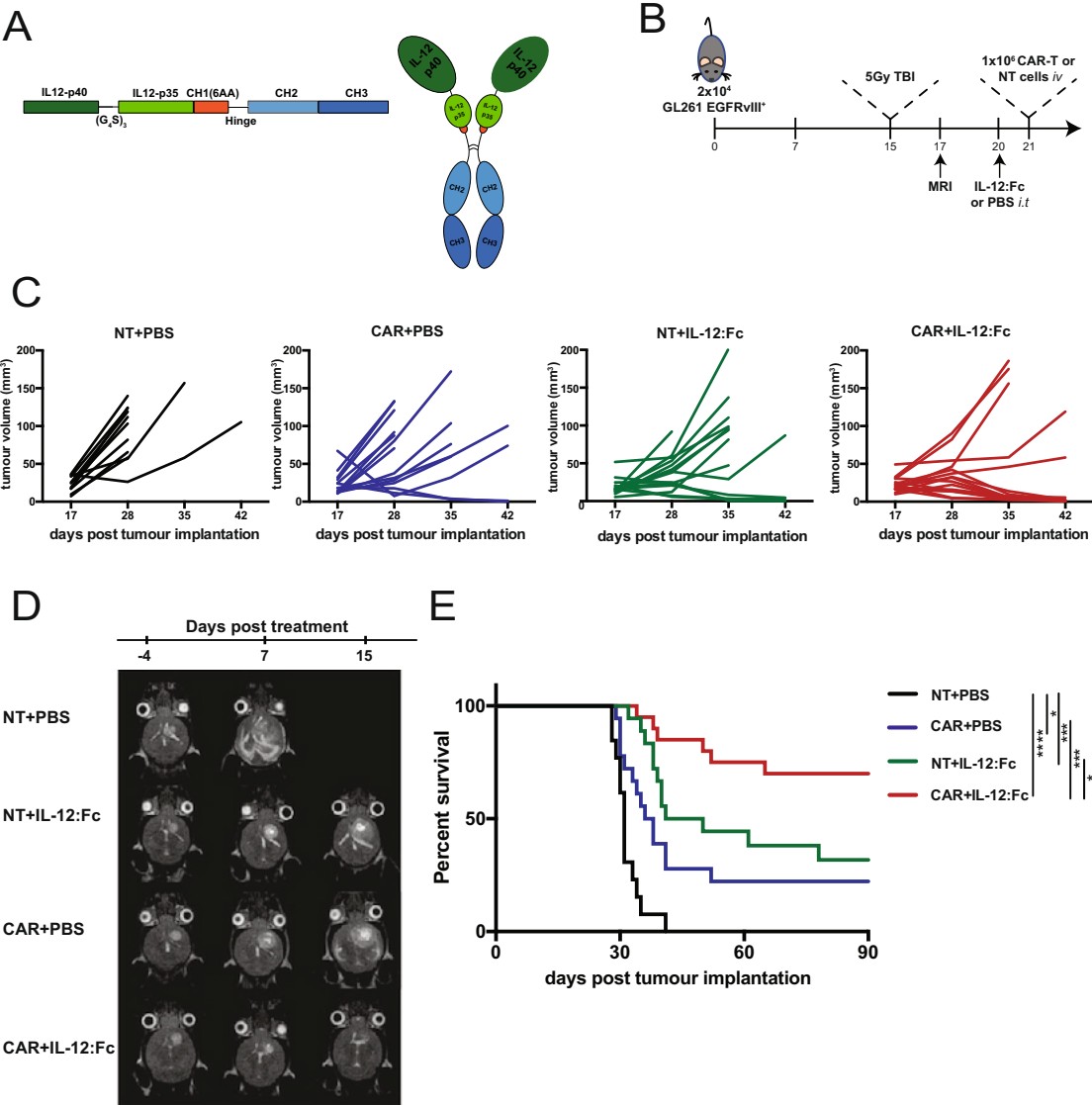

**Fig. 2 Combination of EGFRvIII-specific CAR-T cells and a single dose of locally administered IL-12:Fc results in effective control of late stage orthotopic tumors. A** Scheme of the IL12:Fc construct inserted in the mammalian expression vector pCEP4 and a schematic representation of heterodimeric IL-12 fused to the Fc portion of murine IgG3. **B** Experiment timeline. GL261 EGFRvIII+ cells were implanted in the right striatum at day 0. Mice received 5 Gy TBI on day 15 post implantation, while tumor engraftment was confirmed on day 17. On day 20, mice underwent surgery and received either PBS or 300 ng of IL-12:Fc at tumor site, followed by intravenous injection of $1 \times 10^6$ CAR-T cells or non-transduced cells. Tumor growth was monitored weekly. **C** Tumor volume quantification. **D** Representative MRI images (axial view) of a mouse from each group. **E** Survival curves (NT+PBS $n =$ 14, CAR+PBS $n = 19$, NT+IL-12:Fc $n = 18$, CAR+IL-12:Fc $n = 20$ from four independent experiments, (NT+PBS vs. CAR+PBS $p = 0.0375$, NT+PBS vs. NT+IL-12:Fc $p = 0.0001$, NT+PBS vs. CAR+IL-12:Fc $p < 0.0001$, CAR+PBS vs. CAR+IL-12:Fc $p = 0.0005$, NT+IL-12:Fc vs. CAR+IL-12:Fc $p = 0.0176$, Log-rank test). Source data are provided as a Source Data file.

receptors PD1 and LAG3 and with low levels of IFN-γ and TNF, which was significantly decreased when CAR-T cells are combined with IL-12:Fc (Fig. 3C). The visualization of the cytometry data using dimensionality reduction (tSNE in conjunction with FlowSOM meta-clustering) confirmed the presence of two distinct CAR-T cell clusters defined as LAG3hiPD1hi and LAG3lowPD1low CAR-T cells (Heatmap, Fig. 3D, Supplementary Fig. 3D). In particular, we observed an increased frequency of LAG3lowPD1low CAR-T cells in the combinatorial therapy compared to treatment with CAR-T cells alone (Fig. 3E), which was also confirmed by manual gating (Supplementary Fig. 3E). Of note, the population induced by IL-12 treatment (LAG3lowPD1lowCAR-T cells) showed a higher capability to produce IFN-γ and TNF upon re-stimulation compared to the LAG3hiPD1hi CAR-T population (Fig. 3F). We next evaluated the expression of

other immune checkpoints associated with T cell exhaustion (e.g. TIM3, CD160, CD244, and CD73), whose ligands are expressed on both myeloid and tumor cells[31]. We indeed observed a significant reduction in the expression of all of these additional markers in LAG3lowPD1lowCAR-T cells thus suggesting that this is truly a less exhausted cell population (Fig. 3F). As tumor responses were rapid, longitudinal CAR-T phenotype and function studies were not conducted. Together, these results indicate that IL-12 prevents tumor-infiltrating CAR-T cell dysfunction and promotes the production of CAR-T cell-derived cytokines.

We also performed a detailed analysis of CAR-T cells in the spleen to investigate the contribution of systemic effects of IL-12. t-SNE in combination with FlowSOM metaclustering allowed us to identify two clusters of CAR-T cells defined as IFN-γhi and

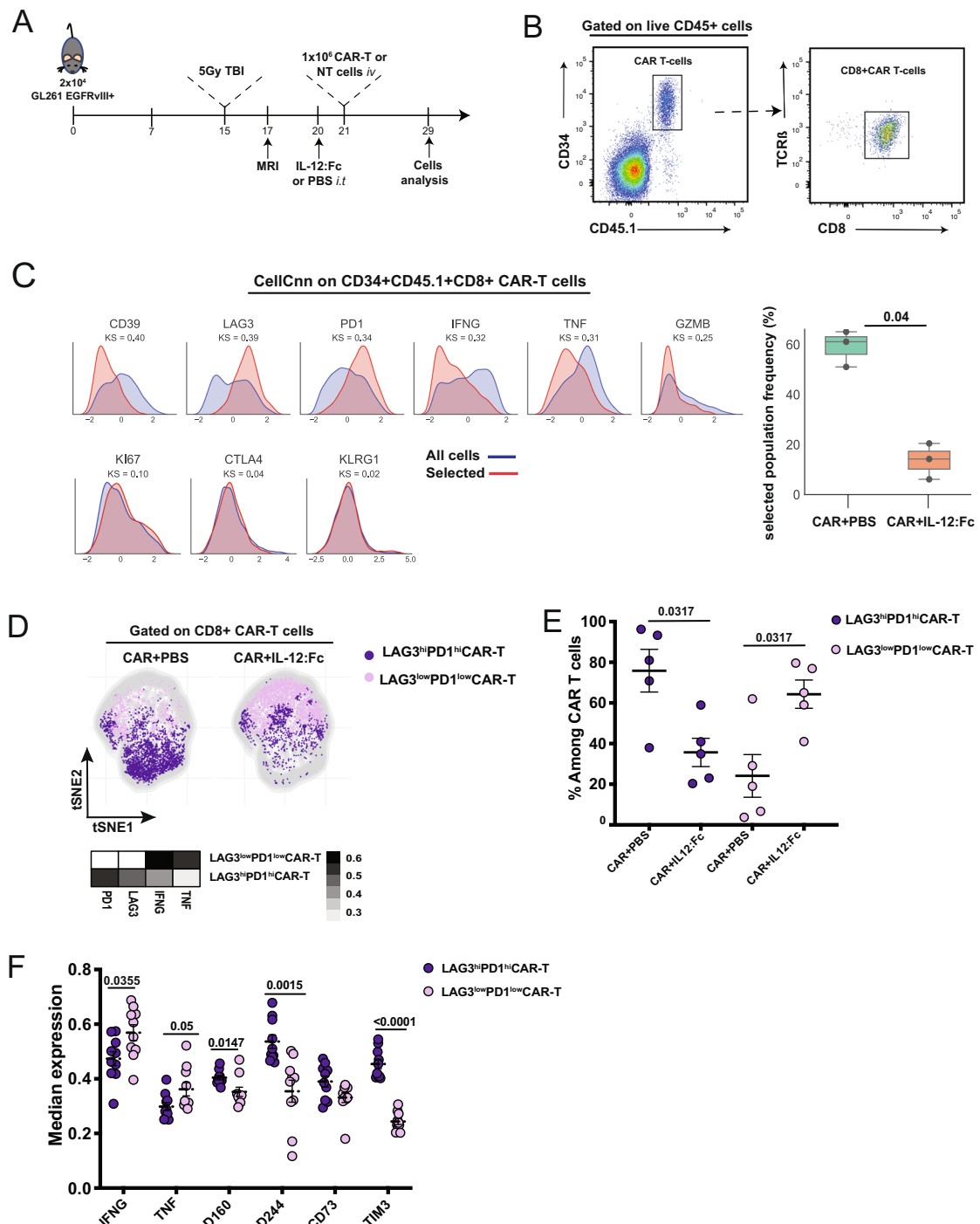

**Fig. 3 Intra-tumoral IL-12:Fc administration reinvigorates EGFRvIII-specific CAR-T cells. A** Experiment timeline. GL261 EGFRvIII+ cells were implanted in the right striatum at day 0. Mice received 5 Gy TBI on day 15 post implantation, while tumor engraftment was confirmed on day 17. On day 20, mice underwent surgery and received either PBS or 300 ng of IL-12:Fc at tumor site, followed by intravenous injection of $1 \times 10^6$ CAR-T cells or non-transduced cells. On day 29 mice were euthanized and FACS analysis was performed on recovered lymphoid cells from brain and spleen. **B** Manual gating of flow cytometry data on CAR-T defined as CD45.1+TCR-β+CD34+CD8+ cells in a representative brain sample among glioma-bearing mice treated with CAR +IL-12:Fc. **C** Relative marker distributions, shown as scaled histograms of arcsinh-transformed marker expression, for all CAR-T cells (violet) and the selected population (red) by CellCNN analysis (left panel); boxplot showing the median frequency and 25th and 75th percentile of the selected population in CAR and CAR+IL-12:Fc group (right panel), $n = 5$ mice per condition, representative of $n = 2$ independent experiments. **D** t-SNE map showing the FlowSOM-guided metaclustering of live intratumoral CD8+CAR-T cells in the different treatment groups; each color represents a metacluster and is associated with a different immune population. Heatmap showing the median marker expression in each metacluster (value range: 0–1, black and white). **E** Frequency of CD8+CAR-T cells for each metacluster in each condition, $n = 5$ mice per condition, representative of $n = 2$ independent experiments. **F** Median expression of selected cell markers shown for LAG3hiPD1hi CAR-T cells and LAG3lowPD1low CAR-T cells in both CAR+PBS and CAR+IL12:Fc conditions, $n = 5$ mice per condition, representative of $n = 2$ independent experiments. Data are presented as mean values ± SEM. 2-tailed unpaired Mann–Whitney $T$ test (**C**, **E**, **F**). Source data are provided as a Source Data file.

IFN-γ[low] CAR-T cells (Heatmap, Supplementary Fig. 3F, G), whose frequency was not affected by the administration of IL-12 (Supplementary Fig. 3H). This reinforces the rationale of administering IL-12 locally at the tumor site to rouse the immunosuppressive TME and to limit the risk of its systemic side effects.

**IL-12 reprograms the endogenous T cell compartment within the glioma TME.** It is well established that IL-12 has the ability to recruit and activate lymphocytes in the TME[13,20]. We examined the recruitment of endogenous T cells to the brain TME by applying tSNE in combination with FlowSOM metaclustering on CD45[+]TCR-β[+]CD34[−] T cells (Supplementary Fig. 4A, B). As shown in Fig. 4, this unbiased analysis identified three clusters of endogenous T cells: CD4[+] T, CD8[+] T and T_reg cells, which were

mainly CD44[+] memory T cells (Heatmap, Fig. 4A). In IL-12-treated gliomas, we observed an increased frequency of CD4[+] T cells and a decrease in frequency and counts of T_reg cells (Fig. 4A, Supplementary Fig. 4C). Importantly, these effects were independent of CAR-T cell treatment. In all three T cell populations, but particularly in CD4[+] T cells, IL-12:Fc led to decreased expression of the checkpoint receptors LAG3 and PD1 (Fig. 4B). CD39, an inhibitor molecule implicated in the adenosine pathway[32], presented variable expression levels in both CD4[+] T cells and CD8[+] T cells (Fig. 4B). However, the ectonucleotidase CD73, which works in cooperation with CD39[33], showed lower expression post combinatorial therapy (Supplementary Fig. 4D). Decreased levels of LAG3 and PD1 in CD8[+] T cells and T_regs were accompanied by an increased production of IFN-γ (Fig. 4B). In the T_reg compartment, high levels of IFN-γ concomitant with a low expression of CD25 and low suppression function may be

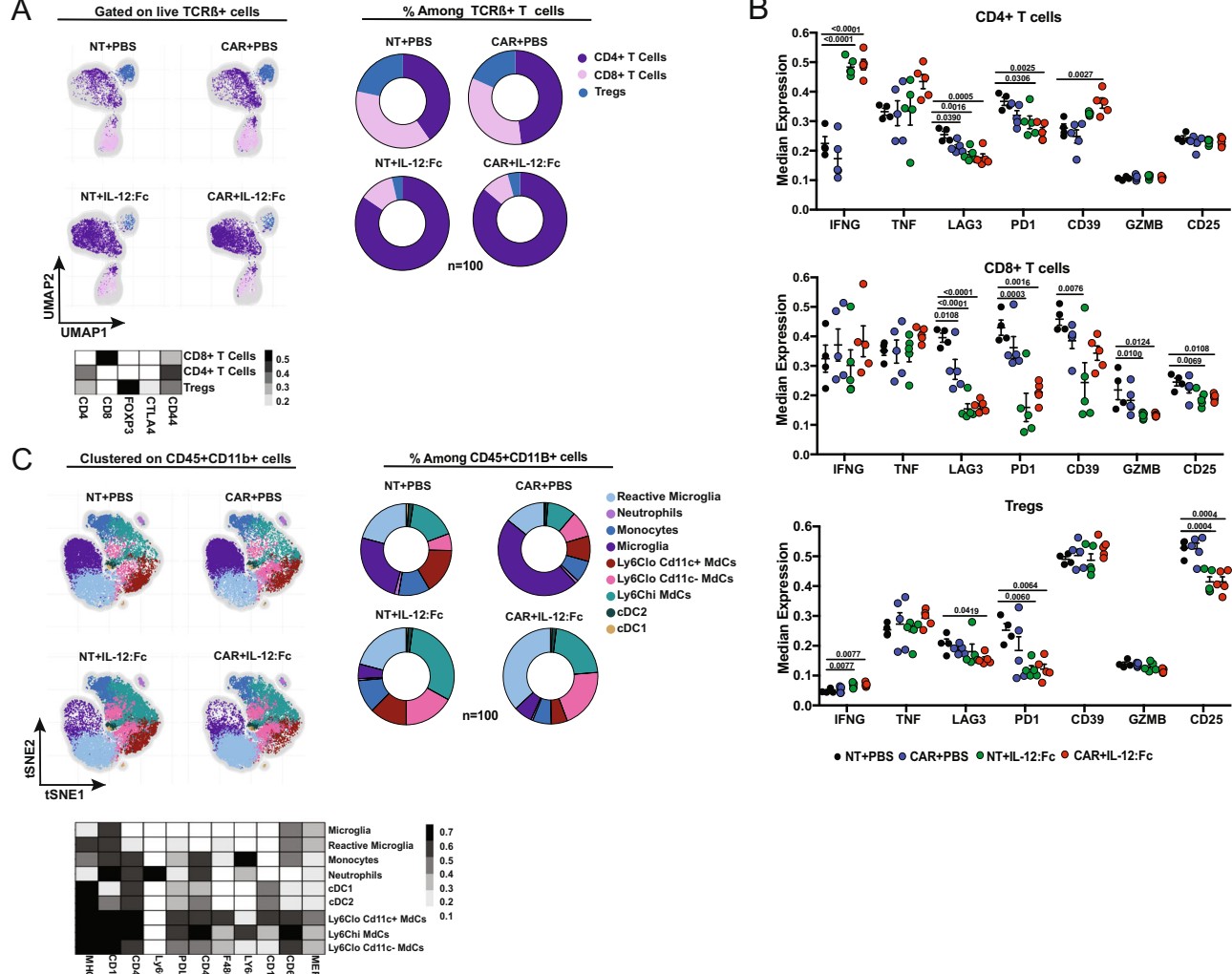

**Fig. 4 IL-12:Fc reshapes the endogenous compartment within the glioma TME. A** Umap showing the FlowSOM-guided metaclustering gated on TCR-β[+] T cells from live CD45[+] cells and heatmap showing the median marker expression for each defined metacluster (value range: 0–1) and Pie charts represent relative frequencies for the three TCR-β[+] T cell subclusters among total TCR-β[+] T cells within the different conditions (NT+PBS n = 4, NT+IL-12:Fc n = −5, CAR+PBS n = 5, CAR+IL-12:Fc n = 5 mice, representative of n = 2 independent experiments). **B** Median expression of selected cell markers shown for all TCR-β[+] T cell subclusters: CD4, CD8, and T_regs, (NT+PBS n = 4, NT+IL-12:Fc n = 5, CAR+PBS n = 5, CAR+IL-12:Fc n = 5 mice, representative of n = 2 independent experiments). **C** t-SNE map showing the FlowSOM-guided metaclustering on CD11b[+]CD45[+] cells and heatmap showing the median marker expression for each defined metacluster (value range: 0–1) and Pie charts represent relative frequencies for the nine CD11b[+]CD45[+] cell subclusters among total CD11b[+]CD45[+] cells within the different conditions, (NT+PBS n = 4, NT+IL-12:Fc n = 5, CAR+PBS n = 5, CAR+IL-12:Fc n = 5 mice, representative of n = 2 independent experiments). Data are presented as mean values ± SEM. Ordinary One-way Anova with Dunnett's multiple comparison test (**B**). Source data are provided as a Source Data file.

indicative of a conversion from "stable" to "fragile" $T_{regs}$[34]. To confirm this notion, we further looked at canonical features associated with $T_{reg}$ suppressive function, such as CD73, ICOS, and GITR. We applied Umap in combination with FlowSOM meta-clustering on $CD4^+FoxP3^+$ T cells exported from the endogenous T cell clusters as the one present in Fig. 4A (Supplementary Fig. 5A). Using this method, we confirmed the presence of two $T_{reg}$ clusters, defined as IFN-$\gamma^{low}$ and IFN-$\gamma^{hi}$ $T_{regs}$ having, respectively, low and high expression of IFN-$\gamma$ (Heatmap, Supplementary Fig. 5A), and verified the increased frequency of IFN-$\gamma^{hi}$ $T_{regs}$ in the presence of IL-12:Fc (Supplementary Fig. 5B). IFN-$\gamma^{hi}$ $T_{regs}$, unlike IFN-$\gamma^{low}$ $T_{regs}$, displayed low expression of CD73 and a trend towards reduced expression of ICOS and OX40, concomitant with an increased production of TNF. This was also associated with a reduced proliferation, as demonstrated by lower Ki67 amounts, consistent with a loss of immunosuppression[35] (Supplementary Fig. 5C). Altogether, these results confirm that intra-tumoral IL-12 is sufficient to increase the ratio of conventional over regulatory T cells, prevent the upregulation of co-inhibitory molecules and sustain IFN-$\gamma$ production in TILs.

**IL-12 reshapes the myeloid compartment within the glioma TME**. We further analyzed whether the improved survival observed with the combinatorial therapy was associated with changes in the myeloid compartment, which represents a significant proportion of the immune cells in the glioma TME[36]. To this end, we performed a multiparametric analysis combined with FlowSOM metaclustering of brain myeloid cells (Fig. 4C, Supplementary Fig. 6A, B). As shown in Fig. 4C, we identified two populations of dendritic cells defined as cDC1 ($CD11c^+CD11b^{lo}$) and cDC2 ($CD11c^+CD11b^{hi}$), three different clusters of monocyte-derived cells (MdCs) ($Ly6C^{hi}$ and $Ly6C^{lo}CD11c^-$ MdCs and $Ly6C^{lo}CD11c^+$ MdCs[37]), two clusters of microglia ($CD11b^{hi}CD45^{lo}MHC$-$II^{lo}$ microglia or $CD11b^{hi}CD45^{lo}MHC$-$II^{hi}$ reactive microglia), one cluster of monocytes ($Ly6C^{hi}MHC$-$II^{lo}$), and one of neutrophils ($CD11b^{hi}Ly6G^+$) (Heatmap, Fig. 4C, Supplementary Fig. 6B). Essentially, in the presence of IL-12, we observed increased frequencies of $Ly6C^{hi}$ and $Ly6C^{lo}CD11c^-$ MdCs and reactive microglia (Fig. 4C, Supplementary Fig. 6C). IL-12:Fc treatment, overall, induced the upregulation of MHC-II on microglial populations, most probably dependent on the IFN-$\gamma$[38] induction (Supplementary Fig. 6D). Similarly, we observed the upregulation of the inhibitory ligand PD-L1 on MdCs in both the IL-12-treated groups (Supplementary Fig. 6D). Moreover, we found that Arginase1 (Arg1), associated with immunosuppressive phagocytes[39], was significantly decreased in $Ly6C^{lo}$ MdCs in the presence of IL-12 (Supplementary Fig. 6E, F). Taken together, these results show IL-12 mediated reshaping of the endogenous T cell and myeloid compartment, which together with the orthogonal effect of IL-12 on CAR-T cell function results in complete eradication of glioma tumors.

**Absence of significant systemic immune-activating effects with local IL-12 delivery**. Given the strong effect observed in tumor control, we sought to verify that the effect of IL-12:Fc remained localized to the intracranial tumor site after injection, because systemic administration of recombinant IL-12 can result in severe adverse effects in humans[40]. We analyzed the serum of treated mice at day 4 and day 11 post IL-12:Fc administration and found no significantly higher levels of IL-12 in the systemic circulation in mice receiving IL-12:Fc compared to controls at these time points (Fig. 5A). However, on day 4 post IL-12:Fc administration, a moderate increase in levels of IL-12-induced cytokines IFN-$\gamma$ (139±73 and 175±59 pg/mL for NT+IL-12:Fc and CAR+IL-12:Fc, respectively) (Fig. 5B) and CXCL9 (103±33 and 134±76 for

NT+IL-12:Fc and CAR+IL-12:Fc, respectively) (Fig. 5C), but no CXCL10 was detected (Fig. 5D). This effect was transient, with levels returned to background at day 11. Moreover, the analysis of serum levels of the pro-inflammatory cytokines IL-6 and GM-CSF showed no differences between the groups (Fig. 5E, F). Taken together, this data indicates that local delivery of IL-12:Fc is accompanied by minimal systemic effects, whilst still providing a significant boost to tumor control.

**Discussion**

Approaches to support CAR-T cell function in the face of an immunosuppressive TME are required to achieve durable anti-tumor immunity against solid tumors, including GBM. Here, we showed in an orthotopic syngeneic model of glioma that a single intra-tumoral injection of IL-12:Fc combined with systemic infusion of CAR-T cells results in the eradication of advanced tumors, whereas either single treatment alone failed to control tumor growth. Combination treatment of CAR-T cells and local IL-12 administration similarly improved efficacy of CAR-T cells in the aggressive and poorly immunogenic B16.F10 tumors implanted intracranially. We demonstrated that the benefit of CAR-T and local IL-12 combination therapy is multi-faceted, as IL-12 administration affects the adoptively transferred CAR-T cells, the endogenous T cell compartment, as well as the myeloid cells in the TME.

In preclinical models of hematologic and solid tumors, IL-12 has been shown to enhance CAR-T cell efficacy[14,16]. Here, we show that within the glioma TME, treatment with IL-12 leads to a decreased proportion of CAR-T cells expressing high levels of PD1 and LAG3, inhibitory receptors which have been associated with reduced functional properties of $CD8^+$ T cells[31]. IL-12 exposure results in an increased proportion of $PD1^{low}LAG3^{low}$ CAR-T cells capable of sustaining their effector function within the TME as demonstrated by their ability to secrete pro-inflammatory cytokines such as IFN-$\gamma$ and TNF upon re-stimulation and downregulate inhibitory receptors associated with T cell dysfunction[41].

Within the endogenous T cell compartment, our results show that IL-12 administration resets the balance between $CD4^+$ effector T cells and $CD4^+$ $T_{reg}$ cells in favor of the $CD4^+$ effector T population (IFN$\gamma^+$). A reduction of $T_{regs}$ has previously been shown to be an important modulatory effect of IL-12 treatment on the TIL compartment in both GL261 and GCS005 preclinical models of glioma[20,42–44]. We observed an increased accumulation of $CD4^+$ effector memory T cells which suggests that IL-12 plays an important role in mediating infiltration of $CD4^+$ T cells. Previous studies have shown that depletion of $CD4^+$ T cells resulted in a complete loss of efficacy of IL-12 treatment[43], indicating that these effector cells play a major role in IL-12-mediated efficacy. A recent study using the GL261 glioma model demonstrated that lack of CD73 improves the efficacy of immune checkpoint therapy with anti-PD1 and anti-CTLA-4[45]. In our system, combination treatment with IL-12 and CAR-T cells decreased the expression of CD73 on $CD4^+$ and $CD8^+$ T cells and hence inverts the ATP/adenosine balance in the TME of late stage experimental glioma[33].

Alongside modification of T cells directly, it has been reported that IL-12 can also skew the myeloid compartment towards a pro-inflammatory phenotype[12,43]. The myeloid compartment plays an important role in GBM immunosuppression and accumulation of immune inhibitory myeloid cells is more prominent in higher tumor grade[36]. We show that IL-12 administration induces an increase of MdCs and reactive microglia in the brain TME of mice treated with combination treatment as compared to mice receiving CAR-T cells only, thus suggesting the ability of

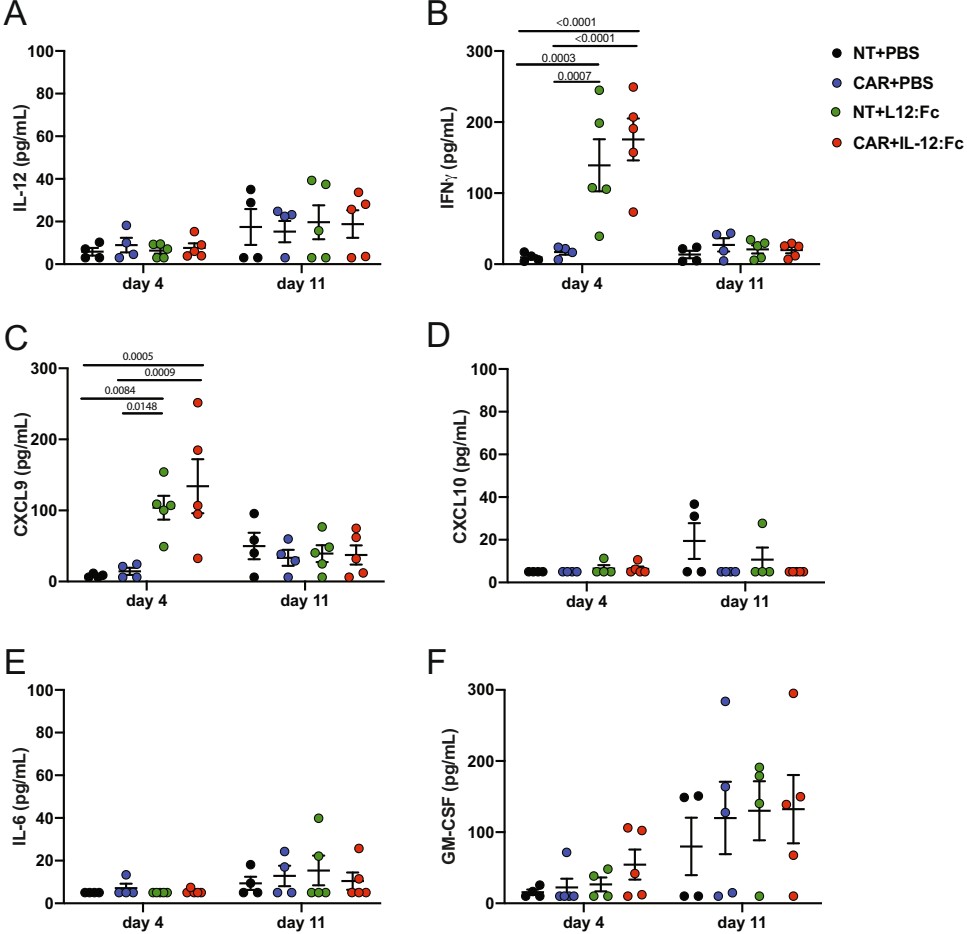

**Fig. 5 Local administration of IL-12:Fc results in minimal systemic effects.** Serum cytokines and chemokines were measured by cytokine bead array (CBA, Legendplex) at day 4 and 11 post IL-12:Fc administration. **A** IL-12, **B** IFN-γ, **C** CXCL9, **D** CXCL10, **E** IL-6, and **F** GM-CSF (NT+PBS $n = 4$, NT+IL-12:Fc $n = 5$, CAR+PBS $n = 4$, CAR+IL-12:Fc $n = 5$ mice from one experiment). Plots show mean ± SEM. Ordinary Two-way Anova with Tukey's multiple comparison test. Source data are provided as a Source Data file.

IL-12 to remodel the myeloid compartment. An increase in MHC class II+ MdCs and microglia potentially supports antigen presentation[46] and may therefore contribute to the observed increased infiltration of endogenous CD4+ T cells. A reinvigoration of the CNS-invading myeloid cells (MdCs) is confirmed by the decrease of Arg1 expression. Arg1 levels correlate with reduced inflammatory capacity in myeloid cells within TME[39,47]. Conversely, the presence of IL-12 led to an upregulation of PD-L1 on MdCs. Therefore, whereas the IL-12-induced PD-L1 expression on MdCs did not lead to tumor evasion, instead it provides a data-driven rationale for combining anti-PD-L1 antibody therapy with CAR-T and IL-12.

While the potential of IL-12 to support anti-tumor immunity has long been recognized, its clinical application has been hampered by its narrow therapeutic index. Transient[48] or regulatable expression[19] in tumor-specific T cells provide alternative approaches to limit IL-12 delivery to the tumor site. However, clinical application of TILs secreting single chain forms of IL-12 (scIL-12) under a NFAT responding promoter showed that this modification did not prevent serum peak levels of IL-12 and IFN-γ, causing significant toxicity[19]. Utilization of direct local injection of a single dose of IL-12:Fc as used in this study allows for the local benefits observed within the tumor microenvironment to be maintained whilst limiting systemic toxicity; we only found a transient upregulation of IL-12-induced cytokines such as IFN-γ and CXCL9, which resolved

11 days post injection. Intracerebral expression of IL-12 was demonstrated to be safe in a phase I feasibility study, where an adenovirus encoding for a drug-inducible IL-12 gene was delivered to the resection cavity[49].

The aim of our work was an immunologic exploration of combining locally administered IL-12 with CAR-T cell therapy. While we believe our findings have clinical relevance, some additional considerations must be made before clinical translation. Firstly, cancer antigen expression can be heterogenous. For instance, in contrast to our model, EGFRvIII is not expressed on all GBM cases and within a tumor its expression is often variable. Furthermore, clinical studies targeting EGFRvIII using vaccine or CAR-T approaches have described antigen loss and tumor escape[10,50,51]. Hence, IL-12:Fc/CAR-T cell therapy for GBM will likely require CAR targeting of multiple antigens[52].

Secondly, in a clinical setting of progressive GBM which has failed standard treatment, high doses of corticosteroids are often administered to reduce brain edema. Corticosteroids could confound immune responses from both CAR-T and IL-12:Fc. In a clinical study in this setting, some patients who are not receiving corticosteroids at trial entry should be included. Recruiting such patients has been feasible in a previous trial of CAR-T cells for recurrent GBM[10]. Notably, a short course of corticosteroids given to treat CAR immunotoxicity does not reduce CAR-T engraftment or activity[53]. Overall, given the poor outcomes following standard therapy in GBM, our anticipation is that successful

immunotherapies will be used earlier in treatment where corticosteroids use is less of a consideration.

In summary, our data show that IL-12 can shape the TME and reprogram it towards a milieu which supports T cell-mediated anti-tumor immunity. While treatment with IL-12 alone is not sufficient to consistently eradicate tumors, combination approaches with checkpoint inhibitors or agonistic antibodies have resulted in complete responses and resistance upon re-challenge in preclinical models[20,43,48]. Clinical application of this approach may however be hampered by limited penetration of checkpoint blocking antibodies across the blood brain barrier and into the tumor bed while incidence of systemic toxicities is significant. Here, we show that single dose intra-tumoral IL-12:Fc is sufficient to sustain CAR-T cell fitness and mediate complete responses in mice with large established gliomas.

Taken together our results support the use of local administration of IL-12 to overcome the barriers encountered by CAR-T cell therapy for GBM so far[10]. This combination approach may create an environment in which CAR-T cells—shown to readily home to local and distant tumor sites[10,54]—can mediate sustained anti-tumor-mediated immunity. We now plan to translate this combination approach of CAR-T cell therapy with single dose intra-tumoral IL-12 administration into a clinical study for patients with GBM.

## Methods

**Cell lines.** GL261 were provided by A. Fontana, Experimental Immunology, University of Zurich, Zurich, Switzerland and cultured in DMEM (Gibco) supplemented with 10% FCS (Biosera), GlutaMAX (Gibco), and sodium pyruvate 1 mM (Gibco). B16.F10 were bought from ATCC and cultured in RPMI (Gibco) supplemented with 10% FCS (Biosera) and GlutaMAX (Gibco). To express EGFRvIII or GD2, cells were transduced in six-well plates with the addition of 3 mL of a γ-retroviral vector in the presence of 10 μg/mL Polybrene (Sigma). The γ-retroviral vector was produced by transient triple transfection of HEK293T using GeneJuice transfection reagent (Merck Millipore) with 4.69 μg of Peq-Pam plasmid (encoding Moloney GagPol), 3.13 μg of VSV-G envelope, and 4.69 μg of retroviral backbone SFG expressing the gene of interest, EGFRvIII or GD2 and GD3 synthase[55] (AddGene plasmid #75013), respectively. Supernatants containing retroviral vector were collected at 48 and 72 h post-transfection and frozen at −80 °C prior to use for GL261 or B16.F10 transduction.

**CAR constructs and transduction of murine T cells.** EGFRvIII_CAR was constructed using the single chain variable fragment (scFv) derived from EGFRvIII-specific MR1 antibody[22], murine CD28-derived transmembrane domain and murine CD28 and CD3ζ intracellular domains. GD2-CAR had the same second-generation CAR structure and contained a scFv derived from murine anti-GD2 antibody K666[27]. For the EGFRvIII-specific CAR, murine CD34 was co-expressed with the CAR to allow detection of CAR-transduced T cells, while for the GD2-specific CAR, Thy1.1 was used as marker gene to generate CAR retroviral particles. Phoenix-Eco (PhEco)-adherent packaging cells (Nolan Laboratory) were transiently transfected with retroviral vectors for the generation of supernatant containing the recombinant retrovirus required for infection of target cells, as previously described[56]. Cells were transfected using Genejuice (Merck Millipore) with 2.68 μg of pCL-eco construct and 4.68 μg of the anti-EGFRvIII or anti-GD2 CAR vector in SFG backbone according to the manufacturers' instructions. Isolated splenocytes were activated with Concanavalin A (Sigma) 2 μg/mL and murine IL-7 (Peprotech) 1 ng/mL for 24 h, then incubated for 72 h with retroviral particles on retronectin-coated (Takara-Bio) 24-well plates (after being spun at 800 × g for 90 min at 32 °C, no brake), in the presence of human IL-2 (100 U/mL; Roche). Transduced cells were injected intravenously into mice 4 days after transduction. CAR-T cells derived from CD45.1 mice were used for all TME analysis experiments by flow cytometry

**In vitro toxicity assay.** Target cells were incubated with $^{51}$Cr (3.7 MBq/1 × 10$^6$ cells) for 1 h at 37 °C in PBS/0.5% BSA. Effector and target cells were incubated at 37 °C for 4 h at 32:1, 16:1, 8:1, and 4:1 ratio, after which supernatant was collected and $^{51}$Cr release was measured with a γ counter and calculated as follows:

[experimental release−background release (targets only)]/[maximum release (Triton X-100)−background release]*100.

**Mice.** All animal studies were performed with the approval of the University College London and UK Home Office and the Swiss Cantonal Veterinary Office of Zurich. Wild-type C57Bl/6J mice (strain code 632) were purchased from Charles River and C57Bl/6 CD45.1 (Ly5.1, strain code 494) mice were a kind gift of Sergio Quezada and were bred at Charles River. Female mice of 6–8 weeks of age were used in all experiments. Mice were housed in a barrier facility in a standard 12 h:12 h light–dark cycle, with food ad libitum. Experimental/control animals were co-housed and spread across cages.

**Animal experiments.** EGFRvIII$^+$ GL261 cells (2 × 10$^4$) or GD2$^+$ B16.F10 (5 × 10$^2$) were stereotactically implanted into the right striatum on day 0 (2 mm right, 1 mm anterior from bregma at a depth of 3 mm), as previously described[20]. Where indicated, mice received 5 Gy TBI using a small animal radiation research platform (SARRP) system (Xstrahl Ltd, UK) to achieve depletion of endogenous lymphocytes prior to CAR-T cell administration[57–59]. CAR-T cells resuspended in 200 μL of PBS were administered at indicated time point by intravenous injection into the tail vein. IL-12:Fc was produced as previously described[20] and stereotactically injected at the same coordinates of tumor injection in 2 μL of PBS.

**Cytokine measurement in serum.** Serum was collected via tail vein bleed. Blood was collected in heparin-coated tubes (Starsted), spun at 2000 × g for 10 min at 4 °C and stored at −80 °C. Cytokine bead array (Biolegend, Legend Plex custom-made) was performed according to manufacturer's protocol and analyzed using Legend Plex software V8.0 (Biolegend).

**Magnetic resonance imaging.** Images were acquired on a small animal 1 T MRI scanner (Bruker ICON) with a 26 mm diameter mouse head coil. Images were acquired using a 2D T2-weighted sequence (TR = 3202 ms, TE = 85 ms, resolution = 0.21 × 0.21 × 0.50 mm$^3$, averages = 15; acquisition time of 6 min), axial orientation. Tumor volumes were exported using Image J (v. 2.0.0-rc-54/1.51 h) and manually calculated using the software ITK Snap (v 3.6.0).

**Survival analysis.** Tumor-bearing animals were monitored weekly by MRI. From day 15 onwards animals were checked daily for neurological symptoms. Animals were euthanized by CO$_2$ followed by cervical dislocation when tumors reached 200 mm$^3$ (as measured by MRI) or when showed symptoms, such as apathy, severe hunchback posture, or weight loss exceeding 20%, whichever appeared first.

**Flowcytometry.** Animals were sacrificed by barbiturate overdose (200 mg/kg) and perfused with cold PBS to remove all circulating leukocytes. Brains were cut into small pieces and incubated with collagenase type IV (0.4 mg/mL) and deoxyribonuclease I (DNase I) (0.2 mg/mL) (Sigma-Aldrich) for 30 min in Hank's balanced salt solution (Thermo Fisher), followed by homogenization through a 19-gauge needle. Tumor infiltrating lymphocytes were enriched by resuspension in Percoll® (GE Healthcare) gradient (30%) and centrifugated (1590 × g at 4 °C, 30 min, brake 1). Cells were then washed twice, blocked using anti-CD32/CD16 (BioLegend) to avoid non-specific binding and stained (Supplementary Table 1).

Spleens were digested as described for the tumors. Erythrocytes were lysed using RBC lysis buffer (NH$_4$Cl 8.3 mg/mL, KHCO$_3$ 1.1 mg/mL, EDTA 0.37 mg/mL). The cells were then spun and resuspended in PBS. Single-cell suspensions were directly used for staining (Supplementary Table 1).

For re-stimulation, cells were incubated for 4 h at 37 °C and 5% CO$_2$ in re-stimulation medium: RPMI 1640 supplemented with 10% FCS and phorbol 12-myristate 13-acetate (50 ng/mL), ionomycin (500 ng/μL), brefeldin A (1 μL/mL; GolgiPlug (BD Biosciences) and Monensin (μL/mL; GolgiStop, BD Biosciences). Cells were then washed twice and stained (Supplementary Table 1). Prior to intracellular staining cells were fixed and permeabilized with fixation/permeabilization solution (Thermo Fisher) for FoxP3 staining or Cytofix/Cytoperm (BD Bioscience) for cytokines staining, according to manufacturer's instructions. Counting beads were added before cells acquisition (AccuCheck Counting Beads, Thermo Fisher). Cells were acquired on a Symphony flow cytometer (BD Biosciences) or Aurora spectral flow cytometer (Cytek). Data were analyzed using FlowJo (version 10.0.8, TreeStar Inc.) and RStudio (Version 3.6.1).

**Histology.** Mice were deeply anesthetized with pentobarbital and transcardially perfused with ice-cold PBS containing 2 mM EDTA, followed by 4% PFA and subsequently embedded in paraffin. 4 μm slides were prepared and stained using a Ventana Discovery XT instrument (Roche), using the Ventana DAB Map detection Kit (760-124). For pre-treatment, either Ventana Protease 3 (equivalent to ficin, 760-2020) and/or Ventana CC1 (950-124), equivalent to EDTA buffer, was used. Slides were haematoxylin counterstained. The following antibodies were used: anti-EGFRvIII (clone L84A, Absolute Antibody) and anti-CD34 (clone RAM34, Thermo Fisher). Secondary antibody (either rabbit anti-mouse or rabbit anti-rat) were from DAKO. Photographs were taken with a Leica DMD108 photographic microscope.

**Statistical analysis.** We utilized unsupervised validated clustering approaches (FlowSOM and CellCNN) to discriminate between different cell populations. For FlowSOM metaclustering, flow cytometer data were compensated and exported with FlowJo software (version 10.0.8, TreeStar Inc.). An unbiased analysis was performed as described previously[30]. To analyze in detail CART cells from brain

and spleen, endogenous T cells, myeloid cells, and endogenous T regs, clusters were exported in RStudio Version 3.6.1 and analyzed using the script available at: https://github.com/BecherLab/High-dimensional-single-cell-Analysis-for-Cytometry-/blob/master/pipeline_FlowSOM.R. The selection of cofactor for data transformation was checked on Cytobank: https://www.cytobank.org/. Statistical analysis was performed using Prism 6.0–8.0 (GraphPad Software, Inc.). Statistical significance of in vivo experiments was determined with a regular one-way ANOVA test with Dunnett's multiple comparison test when every mean was compared to a control mean or Tukey test when every mean was compared with every other mean. Comparison between two groups was performed with Unpaired Mann–Whitney T test. Kaplan–Meier survival analysis was performed to assess survival differences among the treatment groups and P values were calculated with the log-rank test.

**Reporting summary**. Further information on research design is available in the Nature Research Reporting Summary linked to this article.

## Data availability

The flow cytometry data that support the findings are available at: https://data.mendeley.com/datasets/xbvcsdp86v/draft?a=f1a9add1-da74-4033-b215-db3e460f1f00. Source data are provided with this paper.

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

## Acknowledgements

This work was funded by the EU framework 7 consortium ATECT, grant agreement no. 602239 (M.A.P., S.A.Q., B.B.), the Swiss National Science Foundation (grants 733 310030_170320,310030_188450 and CRSII5_183478) (B.B.), the European Union H2020 Project iPC #826121 (B.B.), and a Cancer Research UK Biotherapeutics Drug Discovery Project Award (C22442/A20766) (M.A.P.). K.S. is funded by a Wellcome Trust Clinician Scientist Fellowship and supported by a Great Ormond Street Hospital Children's Charity Clinical Starter Grant (V1287). M.A.P. is supported by the University College London Hospital Biomedical Research Centre. S.A.Q. is funded by a Cancer Research U.K. (CRUK) Senior Cancer Research Fellowship (C36463/A22246) and a CRUK Bio-therapeutic Program Grant (C36463/A20764). A.R.L. is a recipient of a University Research Priority Program (URPP) postdoctoral fellowship. We thank Prof. Sebastian Brandner and Angela Richard-Londt from the UCL IQPath (UCL Institute of Neurology) for their support with histopathology. We thank Dr. Rebecca Carter and Adam Wes-thorpe from Prof. Sharma's lab for their support with the SARRP system. We further thank Mirjam Lutz from the Institute of Experimental Immunology at the University of Zurich for her support with in vivo and post-mortem experiments.

## Author contributions

G.A. tested the CAR constructs, designed and performed experiments, analyzed data and wrote the manuscript. A.R.L. designed and performed flow cytometry assays, analyzed the data and wrote the manuscript. A.H. performed animal experiments and assisted with data interpretation. D.D.F. and N.N. designed flow cytometry assay and assisted with data interpretation. C.L.S performed animal experiments. E.F. assisted with myeloid data interpretation. F.N. performed cytokine bead array analysis. L.R. assisted during the flow cytometry assays. I.P.W. assisted with MRI scans. R.R., T.A.R., and B.M.S. optimized MRI sequences and assisted with MRI scans. T.L.K and M.F.L. assisted in vivo imaging development. S.A.Q. conceived the project and reviewed the manuscript. M.A.P. conceived the project, designed the constructs, reviewed the data and the manuscript. S.T. wrote and reviewed the manuscript. K.S. conceived the project and wrote the manuscript. B.B. conceived the project and reviewed the manuscript.

## Competing interests

B.B. holds shares to Gaeta Therapeutics and a patent on the use of IL-12 tumor targeting in combination with check-point inhibition. The patent is named "COMBINATION MEDICAMENT COMPRISING IL-12 AND AN AGENT FOR BLOCKADE OF T-CELL INHIBITORY MOLECULES FOR TUMOUR THERAPY" with publication number: 20150017121. M.A.P. owns stock, receives salary contribution and research funding from Autolus Ltd. M.A.P. is entitled to a share of royalties earned from patents filed by UCL on his behalf. All other authors declare no competing interests.
