## [Peer Review File · Nature Communications]

REVIEWER COMMENTS

Reviewer #1 (Brain tumor immunology)(Remarks to the Author):

This study attempts to demonstrate the therapeutic benefits of combination therapy with local IL12 and EGFRvIII targeting CAR-T cells in the treatment of glioblastoma using an established glioma murine model. To this effect, the authors report that local delivery of IL-12 combined with CAR-T cell demonstrated control of established murine tumors with increase in infiltration of proinflammatory CD4 T cells, myeloid cells, and decreased T regulatory T cells in the tumor microenvironment. Given the disappointing results thus far of immunotherapy in GBM patients and known global immunosuppression present in GBMs, there is a need to explore the potentials of combination immune-based therapies and better understanding of the effect of these therapies on the tumor microenvironment. IL-12 can directly enhance the activity of effector CD4 and CD8 T cells as well as modulate the activities of tumor infiltrating myeloid cells, and therefore may be a promising adjunct to other immunotherapies such as CAR-T cells in enhancing anti-tumor immune activities.

Major comments:

While the authors' conclusions are reasonable, I have the following concerns:

1. It is unclear the authors' use of the "stress model" of glioblastoma and how it better recapitulates clinical GBM as the CAR-T cells likely will be administered post-surgical resection at initially diagnosis or at recurrence. It may be more beneficial to demonstrate the efficacy of this combination therapy in a less immunogenic model or a GEM model that more accurately recapitulate GBMs. Furthermore, in this study, the mice received whole body irradiation prior to administration of local IL-12 and CAR-T cells. This can potentially confound the distribution and functional capacities of the immune cells. It is unclear the rationale behind the irradiation in these in vivo studies. Another concern of translating this to the clinic is the common use of steroids in the management of GBM patients which has previously been shown to limit the efficacy of immune-based therapies, the authors should comment on the potential effects of steroid use on the efficacy of their proposed therapy regimen.
2. The authors concluded that LAG3 low and PD low CAR-T cells showed increase capacity to produce IFN gamma and granzyme B upon re-stimulation. Though this was measured by intra-cellular staining after stimulation. It may make the conclusions stronger if the authors sorted out the two clusters of CD8 CAR-T cells and then performed functional assay to better quantify these cells' functionality.
3. Would consider adding additional markers of exhaustion such as Tim3, 2B4 etc to better delineate the phenotypes of CAR-T cells in the setting of local IL-12 delivery
4. The authors speculate that increased IFN gamma expression on Tregs with IL-12 management may lead to "fragile"/"unstable" Tregs, it is known that Tregs can upregulate expression of IFN gamma in the setting of IL-12, would consider evaluating for expression of immunosuppressive cytokines in this cell population compared to conventional Tregs or perform Treg suppression assays to confirm the hypothesized decreased suppression of these Tregs.
5. While the authors found upregulation of MHCII on MDCs and a population of microglia with IL12 treatment, there was also up regulation of PD-L1 on these myeloid populations. I believe this deserves further exploration and discussion. As mentioned above, it would be beneficial to include assessment of immunosuppressive cytokines such as IL10, TGF beta or functional assays.
6. Previous studies of CAR-T cells therapy in other solid tumors have demonstrated the concern for the transient nature of immune-modulatory effects by the CAR-T cells on the tumor microenvironment. Can the authors address this particular concern in the context of their proposed combination therapy regimen? May be worthwhile to perform timed sacs post IL12 and CAR-T cell administration to further characterize the temporal changes in TME after therapy.

Minor comments:

1. Would recommend the authors to perform a thorough read through for spelling and grammatical errors.

Reviewer #2 (GBM immunotherapy, IL12)(Remarks to the Author):

Overall this work is interesting demonstrating the benefit of local IL12 expression plus CAR-T cells vs. EGFRvIII using a GBM tumor in a mouse model. Local IL12 in GBM mouse models has been used previously in particular with IL12expressing viruses. Similarly CAR-T cells have been used in an attempt to target GBM. In this paper they put the two strategies together, however, one pitfall is that only one model is used and the model used contains a homogeneous group of cells expressing EGFRvIII. Human GBMs are very heterogeneous and EGFRvIII is expressed at variable levels and not necessarily in all cells. There should be mention of concerns that this specific strategy of eliminating EGFRvIII cells in human GBM will not necessarily achieve a goal of significant improved outcomes as previously shown with a vaccine strategy to EGFRvIII by Sampson et al. (J Clin Oncol. 2010 Nov 1; 28(31): 4722–4729) wherein there was only small beneficial outcome despite recurrences showing loss of EGFRvIII expressing cells in 82% and as noted in follow-up papers by Platten et al (Neuro-Oncology, Volume 19, Issue 11, November 2017, Pages 1425–142) and Weller et al. (Weller et al, Lancet Oncol 2017).

Reviewer #3 (CAR-T in glioma)(Remarks to the Author):

This is a very clear and well-written, scientifically rigorous, novel, and interesting manuscript, and I don't say any of those lightly. The authors demonstrate a relatively simple way to enhance CAR T cell efficacy with an intratumoral injection of soluble IL-12-Fc fusion protein, and demonstrate how this approach does not change CAR T cell infiltration but does maintain function and modify the tumor microenvironment (endogenous T cells, Tregs, and myeloid cells). The immunocompetent orthotopic model is appropriate for these studies. I have only minor changes to suggest:

1. The Discussion should include some comments about the heterogeneity of EGFRvIII expression, and the fact that one limitation of this study is the use of a single murine GBM line that is transduced to express high levels of EGFRvIII.
2. Fig 2 should indicate the route of administration of the CAR T cells (presumably intravenous, but it is not in the text or the legend). It would also be helpful to the reader to include a schematic diagram of how the IL-12:Fc fusion construct design.
3. Fig 4 -- can you clarify if the TCRb+ cells include CAR+ T cells? It would be interesting to specifically do this analysis in the CAR-negative T cells, to isolate the effect to the endogenous T cell compartment.

Marcela Maus

Reviewer 1

(1a). It is unclear the authors' use of the "stress model" of glioblastoma and how it better recapitulates clinical GBM as the CAR-T cells likely will be administered post-surgical resection at initially diagnosis or at recurrence. It may be more beneficial to demonstrate the efficacy of this combination therapy in a less immunogenic model or a GEM model that more accurately recapitulate GBMs.

We thank the reviewer for bringing up these points.

We used an immunocompetent syngeneic orthotopic model of GBM. EGFRvIII CAR-T cell dose was selected whereby CAR-T cells alone have limited efficacy. This approach of "stressing" a model is widely used in the CAR-T field to make incremental improvements¹. We further justify the conditions selected, since clinical application of CAR-T cells in GBM to date rarely results in complete response^{2,3}.

Regards the question of whether our model best recapitulates likely clinical scenarios combined treatment with CAR-T and IL-12 will be used in – we do not yet know what the ideal clinical setting is. In first instance, our approach will most likely be tested at recurrence after failure of standard of care as in O'Rourke et al.² In this setting, the clinical scenario is of progressive disease. A consolidation approach immediately after resection may happen in future. Even in this scenario, often a considerable amount of tumour burden will remain due to limitation in possible resection and invasive parenchymal spread characteristic of this tumour.

Regards use of a GEM model: the absence of a transgenic model with endogenous expression of the CAR target antigen EGFRvIII accessible to us precluded our using such a model. Further, this orthotopic model using GL261 is the most widely used model for development of immunotherapies for GBM⁴⁻⁶ and this well characterized immunocompetent orthotopic model seemed well suited to our goal of systematically exploring a potential synergy between locally administered IL-12 combined with CAR-T cells to achieve tumour eradication.

Regards use of a less immunogenic model: to address this critique and to solidify the synergy between CAR-T cells and locally administered IL-12 in a less immunogenic brain tumour model, we used intracranial implantation of B16.F10 to recapitulate brain metastases of melanoma⁷⁻⁹. B16.F10 cells were transduced to express GD2, a tumour antigen expressed on tumours of neuroectodermal origin including gliomas, melanomas and neuroblastoma.

The B16.F10_GD2 tumour model we generated is a highly aggressive and poorly immunogenic; mice left untreated survive for 15 days only. In this model we showed significantly improved survival in mice receiving combination treatment with IL-12 and CAR-T cells as compared to those treated with either treatment alone. This new data confirming the merit of combined CAR-T cell and local administration IL-12 treatment is described in the Results section and included as **new Supplementary Figure 2**.

(1b) Furthermore, in this study, the mice received whole body irradiation prior to administration of local IL-12 and CAR-T cells. This can potentially confound the distribution and functional capacities of the immune cells. It is unclear the rationale behind the irradiation in these in vivo studies.

We thank the reviewer for raising this point and we regret to not have clarified this better in our first version of the manuscript. Preparative lymphodepletion is required to achieve engraftment and anti-tumour efficacy of adoptively transferred T cells and is now a standard component of CAR-T cell therapy¹⁰. Clinical studies have shown that effective lymphodepletion can be achieved by low dose total body irradiation (TBI) or a combination of fludarabine and cyclophosphamide (Flu/Cy) chemotherapy¹¹. In contrast, low dose cyclophosphamide alone is insufficient to achieve CAR-T cell engraftment¹². Low dose Flu/Cy is difficult to administer in mice. Instead, TBI is the standard reproducible method to achieve

lymphodepletion in mice and has been widely used for this purpose in immunocompetent mouse models^{13,14} including in the context of GBM¹⁵.

While the point that irradiation can confound the distribution and functional capacities of immune cells is well taken, as all treatment groups including the control mice treated with non-transduced T cells without IL-12 received TBI, we demonstrate that the observed anti-tumour immunity of the combination immunotherapy is independent from TBI.

The purpose of use of TBI has now been better explained in the manuscript in the Results section as well as in the Material and Methods section.

(1c) Another concern of translating this to the clinic is the common use of steroids in the management of GBM patients which has previously been shown to limit the efficacy of immune-based therapies, the authors should comment on the potential effects of steroid use on the efficacy of their proposed therapy regimen.

The reviewer is correct – patients with GBM often receive high doses of corticosteroids. Interestingly, high doses of corticosteroids given to treat immune toxicity in the CD19 CAR-T cell setting do not affect CAR T-cell function and clinical outcome¹⁶. However, in a clinical study testing our approach, we would select patients who are not receiving corticosteroids at trial entry and use a short course of steroids if required to manage toxicity after CAR-T cells. We appreciate for patients with relapsed / progressive disease who are typically recruited to early CAR-T cell studies, this may be difficult; but this has already been shown to be feasible in a previous trial of CAR-T cells for recurrent GBM¹⁷. Importantly, if such treatments are shown to be effective in end-stage patients, we anticipate that these approaches will be used earlier in treatment lines where administration of corticosteroids becomes less of a consideration. We very much value this comment as a relevant clinical consideration and have added to the discussion of the manuscript.

(2). The authors concluded that LAG3 low and PD low CAR-T cells showed increase capacity to produce IFN gamma and granzyme B upon re-stimulation. Though this was measured by intracellular staining after stimulation. It may make the conclusions stronger if the authors sorted out the two clusters of CD8 CAR-T cells and then performed functional assay to better quantify these cells' functionality

We have attempted to recover sufficient CAR-T cells from treated mice to perform ex vivo functional experiments. However, as shown in **Supplementary fig. 3**, the number of CAR-T cells recovered from the brain are low, and due to the well-recognised difficulty of culturing murine T cells ex vivo – in particular after a sorting step - conducting these experiments proved not possible. The capacity of IL-12 to increase cytotoxicity in vitro is well known^{18,19}. Here we demonstrate the enhanced in vivo cytotoxicity of CAR-T cells when combined with intratumoural administration of IL-12.

(3). Would consider adding additional markers of exhaustion such as Tim3, 2B4 etc to better delineate the phenotypes of CAR-T cells in the setting of local IL-12 delivery

As suggested, we have further studied the phenotype of the CAR-T cells upon exposure to IL-12 using an expanded panel of activation and exhaustion markers. We observed that CAR-T cells low for PD1 and LAG3 not only produced high amounts of IFN- γ but also had increased production of TNF and downregulated expression of exhaustion markers such as TIM3, CD244 (2B4) and CD160. This new data is described in the Results section and shown in **new Figure 3F**.

(4). The authors speculate that increased IFN gamma expression on Tregs with IL-12 management may lead to "fragile"/"unstable" Tregs, it is known that Tregs can upregulate expression of IFN gamma in the setting of IL-12, would consider evaluating for expression of immunosuppressive cytokines in this cell population compared to conventional Tregs or perform Treg suppression assays to confirm the hypothesized decreased suppression of these Tregs.

This is a very interesting point. As mentioned above, ex vivo experiments with T_{regs} recovered from the CNS turned out to be impossible or would require an extraordinary and prohibitive number of mice. Instead, to address this question, we quantified T_{reg} fragility measuring canonical markers associated with T_{reg} suppression. We show that the frequency of T_{regs} able to produce high amounts of IFN- γ

increases in the presence of IL-12 (**new Supplementary Fig. 5B**) and also, similar to what others previously found²⁰, that these cells present a less suppressive phenotype associated with a decreased expression of CD73, low proliferation and an increased production of IFN- γ and TNF (**new Supplementary Fig. 5C**). We could not additionally measure suppressive cytokines such as IL-10 via FACS, due to the mutually exclusive staining conditions required for Foxp3 and IL-10.

(5). While the authors found upregulation of MHCII on MDCs and a population of microglia with IL12 treatment, there was also up regulation of PD-L1 on these myeloid populations. I believe this deserves further exploration and discussion. As mentioned above, it would be beneficial to include assessment of immunosuppressive cytokines such as IL10, TGF beta or functional assays.

As requested, we measured TGF beta and IL-10 in brain cell lysate which gives us a global picture of the cytokine milieu present in glioma and found no significant differences. We then measured the immunosuppressive enzyme, Arginase¹²¹ by flow cytometry and found this enzyme was significantly downregulated in the IL-12 conditions in MDCs confirming a reinvigoration of the myeloid compartment in the presence of IL-12. This new data is shown in **new Supplementary Fig. 6E**.

As reported previously the expression of PD-L1 on myeloid cells can be induced by lymphocyte-derived IFN- γ ²² and it has already been shown that tumour and non-tumour cells upregulate PD-L1 after treatment with IL-12 at the tumour site²³. We showed that the main PD-L1 ligand (PD1) is substantially reduced on both the endogenous compartment and adoptively transferred cells and that the combinatorial therapy of CAR-T plus IL-12 retains anti-tumour efficacy despite the upregulation of PD-L1 on myeloid cells. The reviewer however raises an interesting point here and the data suggest that combining CAR-T cells with intratumoural IL-12 and anti-PD-L1 would be an attractive data driven-therapy and we expanded the discussion accordingly.

(6). Previous studies of CAR-T cells therapy in other solid tumors have demonstrated the concern for the transient nature of immune-modulatory effects by the CAR-T cells on the tumor microenvironment. Can the authors address this particular concern in the context of their proposed combination therapy regiment? May be worthwhile to perform timed sacs post IL12 and CAR-T cell administration to further characterize the temporal changes in TME after therapy.

This is a very astute point and an excellent suggestion. Our data show that the intratumoural application of IL-12 converts the TME and makes it conducive to CAR-T cell immunotherapy. We argue that priming the TME is essential for CAR-T cell function as opposed to the length of effect of IL-12 on the TME. In our model, tumours are cleared within 2 weeks from treatment administration, making the analysis at different time points virtually impossible. It is not yet known what the temporal requirements are for CAR T-cell therapy in GBM. In the setting of a different solid tumour diffuse large B-cell lymphoma clinical experience has shown these are rapidly cleared by CART cell therapy and that only a short period of CAR-T cell activity is required for sustained responses¹⁶. This clinical experience suggests that our observed rapid effects may not be an unrealistic prediction of the clinical scenario. We have now explained in the Results section, that as tumour responses were rapid longitudinal immunological studies were not conducted.

Minor comments:

perform a thorough read through for spelling and grammatical errors.

We have performed a careful check of spelling and grammar.

Reviewer 2.

Only one model is used and the model used contains a homogeneous group of cells expressing EGFRvIII. Human GBMs are very heterogeneous and EGFRvIII is expressed at variable levels and not necessarily in all cells. There should be mention of concerns that this specific strategy of eliminating EGFRvIII cells in human GBM will not necessarily achieve a goal of significant improved outcomes as previously shown with a vaccine strategy to EGFRvIII by Sampson et al. (J Clin Oncol. 2010 Nov 1; 28(31): 4722–4729) wherein there was only small beneficial outcome despite recurrences showing loss of EGFRvIII expressing cells in 82% and as noted in follow-up papers by Platten et al (Neuro-Oncology, Volume 19, Issue 11, November 2017, Pages 1425–142) and Weller et al. (Weller et al, Lancet Oncol 2017).

This concern is well noted and EGFRvIII negative escape has indeed been described after treatment with EGFRvIII targeted CAR-T cell therapy as well as vaccine approaches as highlighted by the reviewer. In our study we used EGFRvIII as it provides a tumour-specific antigen that conveniently crosses mouse and human barrier. We acknowledge that EGFRvIII is not expressed on all GBM tumour cells and that level of expression of EGFRvIII is often heterogenous. In clinical development of CAR-T cell therapy for GBM we anticipate targeting more than one antigen²⁴. Here however, we used EGFRvIII (and GD2 for the newly included B16.F10 model) to provide a preclinical model system that allows us to focus on exploration of CAR-T cell therapy in combination with the intratumoural application of IL12. We have now included these considerations and references in the discussion of the manuscript.

“The aim of our work was an immunologic exploration of combining locally administered IL-12 with CAR-T cell therapy. While we believe our findings have clinical relevance, some additional considerations must be made before clinical translation. Firstly, cancer antigen expression can be heterogenous. For instance, in contrast to our model, EGFRvIII is not expressed on all GBM cases and within a tumor its expression is often variable. Furthermore, clinical studies targeting EGFRvIII using vaccine or CAR-T approaches have described antigen loss^{2,25,26}. Hence, IL-12-Fc/CAR-T cell therapy for GBM will likely require CAR targeting of multiple antigens²⁴.”

Reviewer 3

1. The Discussion should include some comments about the heterogeneity of EGFRvIII expression, and the fact that one limitation of this study is the use of a single murine GBM line that is transduced to express high levels of EGFRvIII.

We agree and are thankful for the suggestion (Please also see our response to reviewer 2). These limitations are well noted and are now included in the discussion of the manuscript as quoted below.

“The aim of our work was an immunologic exploration of combining locally administered IL-12 with CAR-T cell therapy. While we believe our findings have clinical relevance, some additional considerations must be made before clinical translation. Firstly, cancer antigen expression can be heterogenous. For instance, in contrast to our model, EGFRvIII is not expressed on all GBM cases and within a tumor its expression is often variable. Furthermore, clinical studies targeting EGFRvIII using vaccine or CAR-T approaches have described antigen loss^{2,25,26}. Hence, IL-12-Fc/CAR-T cell therapy for GBM will likely require CAR targeting of multiple antigens²⁴.”

In addition, to demonstrate the synergy between CAR-T cells and locally administered IL-12 in an aggressive and poorly immunogenic tumour, we used intracranial implantation of B16.F10 which recapitulates brain metastases of melanoma. B16.F10 cells were transduced to express GD2, a tumour antigen expressed on tumours of neuroectodermal origin including gliomas, melanomas and neuroblastoma. In this model we showed significantly improved survival in mice receiving combination treatment with IL-12:Fc and CAR-T cells as compared to those treated with either treatment alone. This

new data is described in the second paragraph of the Results section and included as **new Supplementary Figure 2**.

2. Fig 2 should indicate the route of administration of the CAR T cells (presumably intravenous, but it is not in the text or the legend). It would also be helpful to the reader to include a schematic diagram of the IL-12:Fc fusion construct design.

The administration route for the CAR-T cells was intravenous via the tail vein. This information is now included in the 'animal experiments' section of Materials and Methods and the legends of Figure 1, 2, and 3. A schematic of the IL-12:Fc design has now been included as **new Figure 2A**.

3. Fig 4 -- can you clarify if the TCRb+ cells include CAR+ T cells? It would be interesting to specifically do this analysis in the CAR-negative T cells, to isolate the effect to the endogenous T cell compartment.

We apologise for not having included the required detail in the description of our methodology. This is now further clarified in the manuscript. To analyse the effect of IL-12 plus CAR-T on the endogenous compartment we selectively excluded the adoptively transferred cells from the TCR beta positive T cells through a specific gating strategy. For analysis of the endogenous T cells, we first gated on cells that were positive for CD45 and negative for CD11B. Then we selected T cells positive for both CD45 and TCR beta and among these cells we excluded CAR-T (co-expressed marker CD34 positive) cells drawing a gate around cells that were only positive for TCR beta but negative for CD34. This gating strategy is now shown in **new Supplementary fig. 4A** and additional detail is now included in the Results section and in the Material and Methods section.

REFERENCES

1. Zhao, Z. *et al.* Structural Design of Engineered Costimulation Determines Tumor Rejection Kinetics and Persistence of CAR T Cells. *Cancer Cell* **28**, 415–428 (2015).
2. O'Rourke, D. M. *et al.* A single dose of peripherally infused EGFRvIII-directed CAR T cells mediates antigen loss and induces adaptive resistance in patients with recurrent glioblastoma. *Sci. Transl. Med.* **9**, (2017).
3. Brown, C. E. *et al.* Regression of Glioblastoma after Chimeric Antigen Receptor T-Cell Therapy. *N. Engl. J. Med.* **375**, 2561–2569 (2016).
4. Kim, J. E. *et al.* Combination therapy with anti-PD-1, anti-TIM-3, and focal radiation results in regression of murine gliomas. *Clin. Cancer Res.* **23**, 124–136 (2016).
5. Weiss, T. *et al.* NKG2D-Dependent antitumor effects of chemotherapy and radiotherapy against glioblastoma. *Clin. Cancer Res.* **24**, 882–895 (2018).
6. Pituch, K. C. *et al.* Adoptive Transfer of IL13R α 2-Specific Chimeric Antigen Receptor T Cells Creates a Pro-inflammatory Environment in Glioblastoma. *Mol. Ther.* **26**, 986–995 (2018).
7. Vom Berg, J. *et al.* Intratumoral IL-12 combined with CTLA-4 blockade elicits T cell-mediated glioma rejection. *J. Exp. Med.* **210**, 2803–11 (2013).
8. Wainwright, D. A. *et al.* Durable therapeutic efficacy utilizing combinatorial blockade against IDO, CTLA-4, and PD-L1 in mice with brain tumors. *Clin. Cancer Res.* **20**, 5290–5301 (2014).
9. Bridle, B. W. *et al.* Immunotherapy Can Reject Intracranial Tumor Cells without Damaging the Brain despite Sharing the Target Antigen. *J. Immunol.* **184**, 4269–4275 (2010).
10. Gattinoni, L. *et al.* Removal of homeostatic cytokine sinks by lymphodepletion enhances the efficacy of adoptively transferred tumor-specific CD8⁺ T cells. *J. Exp. Med.* **202**, 907–912 (2005).

11. Dudley, M. E. *et al.* Adoptive cell therapy for patients with metastatic melanoma: Evaluation of intensive myeloablative chemoradiation preparative regimens. *J. Clin. Oncol.* **26**, 5233–5239 (2008).
12. Turtle, C. J. *et al.* Immunotherapy of non-Hodgkin's lymphoma with a defined ratio of CD8 + and CD4 + CD19-specific chimeric antigen receptor–modified T cells. *Sci. Transl. Med.* **8**, 355ra116-355ra116 (2016).
13. Slaney, C. Y. *et al.* Dual-specific chimeric antigen receptor T cells and an indirect vaccine eradicate a variety of large solid tumors in an immunocompetent, self-antigen setting. *Clin. Cancer Res.* **23**, 2478–2490 (2017).
14. Mardiana, S. *et al.* A multifunctional role for adjuvant anti-4-1BB therapy in augmenting antitumor response by chimeric antigen receptor T cells. *Cancer Res.* **77**, 1296–1309 (2017).
15. Sampson, J. H. *et al.* EGFRvIII mCAR-modified T-cell therapy cures mice with established intracerebral glioma and generates host immunity against tumor-antigen loss. *Clin. Cancer Res.* **20**, 972–984 (2014).
16. Neelapu, S. S. *et al.* Axicabtagene ciloleucel CAR T-cell therapy in refractory large B-Cell lymphoma. *N. Engl. J. Med.* **377**, 2531–2544 (2017).
17. O'Rourke, D. M. *et al.* A single dose of peripherally infused EGFRvIII-directed CAR T cells mediates antigen loss and induces adaptive resistance in patients with recurrent glioblastoma. *Sci. Transl. Med.* **9**, eaaa0984 (2017).
18. Yang, Y. G. & Sykes, M. The role of interleukin-12 in preserving the graft-versus-leukemia effect of allogeneic CD8 T cells independently of GVHD. *Leuk. Lymphoma* **33**, 409–420 (1999).
19. Tugues, S. *et al.* New insights into IL-12-mediated tumor suppression. *Cell Death Differ.* **22**, 237–246 (2015).
20. Cao, X. *et al.* Interleukin 12 stimulates IFN- γ -mediated inhibition of tumor-induced regulatory T-cell proliferation and enhances tumor clearance. *Cancer Res.* **69**, 8700–8709 (2009).
21. Zhang, I. *et al.* Characterization of arginase expression in glioma-associated microglia and macrophages. *PLoS One* **11**, 1–16 (2016).
22. Chen, R. Q., Liu, F., Qiu, X. Y. & Chen, X. Q. The prognostic and therapeutic value of PD-L1 in glioma. *Front. Pharmacol.* **9**, (2019).
23. Fallon, J. K., Vandever, A. J., Schlom, J. & Greiner, J. W. Enhanced antitumor effects by combining an IL-12/anti-DNA fusion protein with avelumab, an anti-PD-L1 antibody. *Oncotarget* **8**, 20558–20571 (2017).
24. Hegde, M. *et al.* Tandem CAR T cells targeting HER2 and IL13R α 2 mitigate tumor antigen escape Find the latest version: Tandem CAR T cells targeting HER2 and IL13R α 2 mitigate tumor antigen escape. *J. Clin. Invest.* **126**, 3036–3052 (2016).
25. Sampson, J. H. *et al.* Immunologic escape after prolonged progression-free survival with epidermal growth factor receptor variant III peptide vaccination in patients with newly diagnosed glioblastoma. *J. Clin. Oncol.* **28**, 4722–4729 (2010).
26. Platten, M. EGFRvIII vaccine in glioblastoma-InACT-IVe or not ReACTive enough? *Neuro. Oncol.* **19**, 1425–1426 (2017).

REVIEWERS' COMMENTS

Reviewer #1 (Remarks to the Author):

concerns have been addressed.

Reviewer #2 (Remarks to the Author):

The authors have sufficiently addressed my initial concerns.

Reviewer #3 (Remarks to the Author):

the authors have addressed my concerns